# Co-Application of Silver Nanoparticles and Symbiotic Fungus *Piriformospora indica* Improves Secondary Metabolite Production in Black Rice

**DOI:** 10.3390/jof9020260

**Published:** 2023-02-15

**Authors:** Shikha Solanki, G. B. V. S. Lakshmi, Tarun Dhiman, Samta Gupta, Pratima R. Solanki, Rupam Kapoor, Ajit Varma

**Affiliations:** 1Amity Institute of Microbial Technology, Amity University, Sector-125, Noida 201303, India; 2Special Centre for Nanoscience, Jawaharlal Nehru University, New Delhi 110067, India; 3Department of Botany, University of Delhi, New Delhi 110007, India

**Keywords:** *Serendipita indica*, endosymbiont, nano-embedded fungus, confocal microscopy, scanning electron microscopy, *Oryza sativa* L. *indica*, nano-bioformulation

## Abstract

In the current research, unique Nano-Embedded Fungus (NEF), made by the synergic association of silver nanoparticles (AgNPs) and endophytic fungus (*Piriformospora indica*), is studied, and the impact of NEF on black rice secondary metabolites is reported. AgNPs were synthesized by chemical reduction process using the temperature-dependent method and characterized for morphological and structural features through UV visible absorption spectroscopy, zeta potential, XRD, SEM-EDX, and FTIR spectroscopy. The NEF, prepared by optimizing the AgNPs concentration (300 ppm) in agar and broth media, showed better fungal biomass, colony diameter, spore count, and spore size than the control *P. indica*. Treatment with AgNPs, *P. indica*, and NEF resulted in growth enhancement in black rice. NEF and AgNPs stimulated the production of secondary metabolites in its leaves. The concentrations of chlorophyll, carotenoids, flavonoids, and terpenoids were increased in plants inoculated with *P. indica* and AgNPs. The findings of the study highlight the synergistic effect of AgNPs and the fungal symbionts in augmenting the secondary metabolites in leaves of black rice.

## 1. Introduction

The evolution of nanotechnology has strengthened the effective use of nanoparticles and has emerged as a radical tool for enhancing agricultural practices and moving to a more sustainable and resilient farming sector [1,2,3]. Nanoscale Particles (NSPs) are atomic or molecular-sized particles of dimension between 1 nm and 100 nm that can modify their physiochemical properties compared to bulk material [4,5]. Agricultural nanotechnology has presented a wide range of possibilities to utilize nanoparticles to enrich crops and increase their productivity. It has the potential to revolutionize the agro sector and its allied fields through its varied applications [6,7]. There is a wide range of nanoparticles conferring plant growth-promoting abilities, providing diverse implications in the field of agriculture. In recent studies, FeO, ZnO, and Zn-Cu-Fe-oxide nanoparticles have shown a positive growth response to the seedlings of mung (*Vigna radiata*) [8]. The use of TiO_2_ nanoparticles has improved the metabolism of nitrogen as well as photosynthetic activity in spinach plants while enhancing drought tolerance capacity in wheat, along with the increase in the content of starch and gluten present in its seed [9,10,11]. Nanoparticles of silver, iron, and carbon were evaluated and demonstrated growth-promoting capabilities such as stress tolerance, increased yield, early germination, and increased flowering rate in plants [12,13].

Silver is a naturally occurring element. Silver in any form, to a certain optimum concentration, is non-toxic to the human immune system as well as the cardiovascular, nervous, and metabolic systems. It is a good dietary source and helps in crop fortification [14]. Silver deficiency is one of the major limiting factors responsible for the low productivity and yield of crops. Nanotechnology has enabled the intensified use of AgNPs in enhancing crop productivity. AgNPs can be synthesized utilizing the chemical reduction method and do not require an expensive servo control system [15,16,17]. The development of AgNPs have restored interest in their positive microbial effects since the widespread use of modern mycology studies. The interaction of AgNPs with fungal symbionts is called Nano-Embedded Fungus (NEF). NEF formulation has been a focal point of study in agro nanotechnology in recent times. Few nanoparticles other than AgNPs along with endophytic fungus have been used to improve agricultural productivity [18].

*Piriformospora indica*, lately also referred to as *Serendipita indica*, acts as a biofertilizer, phyto-remediator, regulator of metabolic activities, and biological herbicide, as well as a bio-pesticide that promotes growth in plants [19,20]. It is a facultative root endophytic fungus belonging to the Basidiomycota order *Sebacinales*, involved in association with a wide range of plant species [21]. It can enhance the growth and yield of both monocotyledons and dicotyledonous plants by colonizing their roots and enhancing plants’ resistance to biotic and abiotic stresses [21]. Due to these positive effect on plants and their ability to grow in axenic culture, *Pir. indica* has become a unique model fungus to study the molecular and physiological basis of various symbiotic plant–microbe interactions. It also helps in plant fortification due to its diverse properties as a bio-fertilizer and bio-control agent.

This research aims to evolve a nanotechnology-augmented fungal endosymbiont in order to enhance the production of secondary metabolites in black rice (*Oryza sativa* L.) leaves. Black rice is the second most consumed grain and one of the main crops that provide food and energy for over half of the world’s population [22]. Due to its effectiveness in maintaining good health, it is consumed as a functional food. It has a wide range of bioactive and nutritive elements including metabolites such as phenols, flavonoids, terpenes, steroids, alkaloids, and carotenoids, which act either as defensive agents or as plant growth regulators [23]. Black rice, which has been cultivated for more than 4000 years, first appeared in China and, at present, is consumed as functional food across the world, especially in Asian countries [24]. Black rice is a superfood of the 20th century, according to research. The fact that it extends life has earned it the nickname “long-life rice” because of its high nutritional value. In addition, it can potentially be utilized to make nutritious foods and drinks, such as gluten-free cereals, giving consumers additional health benefits [25]. *P. indica* enhances biomass production and is tolerant to numerous biotic and abiotic stresses. This research assesses the effect of AgNPs on black rice in the presence of endophytic fungi—*P. indica.* It seeks to develop a nanotechnology-embedded endosymbiont that enhances secondary metabolite production and can provide innovative solutions to the challenges that conventional farming practices cannot address.

## 2. Methods

### 2.1. Materials and Reagents

All the analytical reagents, such as silver nitrate, sodium borohydrate PVP Polyvinyl pyrrolidine, and nutrient agar, were purchased from SRL labs. Organic jaggery was obtained from Dhampur. Root endophyte of *Piriformospora indica* was discovered, screened, and identified as an endophyte from orchid plants of the Thar deserts in Rajasthan, India, by Prof. Ajit Varma and the research group (Verma et al. (1998) [26] and maintained at Amity Institute of Microbial Technology (AIMT) as accession number DSM 11827. The pure inoculum was obtained from the (AIMT) laboratory and maintained in 4% jaggery media following the methodology of). Attri et al. (2018) [27] Grains of the black rice variety Chakhao Poireiton were obtained from Manipur, Northeast India. Milli-Q water was used in all the Sconducted experiments. Before use in experiments, the glass wares were washed, autoclaved, and dried rinsed with Milli-Q water.

### 2.2. Synthesis and Characterization of AgNPs

The AgNPs were synthesized using the temperature-dependent chemical reduction method [28]. Briefly, 20 mL of 1.5 mM silver nitrate was taken in the burette at room temperature. Then, 300 mL of 2.0 mM sodium borohydrate was taken in the beaker and stirred in an ice bath. After the solution was completely cooled, silver nitrate was added to it drop by drop. Once, the color change was observed, 16 drops of 0.3% PVP were added under continuous stirring for 5 min. In this process of synthesis, the sharp SPR peak was recorded by the UV visible spectroscopy (UV-Vis) to confirm the formation of AgNPs. The reduction of Ag ions occurred when the silver nitrate solution was treated with sodium borohydrate. Characterization of chemically synthesized AgNPs was performed by X-ray Diffraction (XRD), UV visible (UV-Vis) absorption spectroscopy, Zeta potential, Scanning Electron Microscopy–Energy Dispersive X-ray analysis (SEM–EDX), and Fourier transform infrared resonance spectroscopy (FTIR). The mycelial morphology, cellular ultrastructure, and internal machinery were examined using SEM and confocal microscopy techniques.

X-ray Diffraction analysis was performed, and Rigaku MiniFlex600 X-ray Diffractometer Riga, Lativa was used with Cu Kα radiation at wavelength λ = 1.54 A° operating at 40 kV and 15 mA. The XRD pattern of the sample silver nanoparticles was obtained in the 2θ range of 10–90° at a scan rate of 4° per minute with a 0.02-degree step size. The UV visible absorption spectrum was obtained with a 1 cm path length quartz cell using a T90+ UV/VIS Spectrometer by PG Instruments Limited Leicestershire, United Kingdom. Perkin-Elmer Spectrum T machine Shelton, USA was used to carry out the FTIR spectrum. Zeta Potential Analyser ZEE COM Microtech Co., Limited, Takidai, Japan was used to study the zeta potential of silver nanoparticles. The scanning electron microscope (EVO-18, Zeiss, Jena, Germany), at the Amity Institute of Advanced Research and Studies-Materials and Devices (AIARS-M&D), Amity University, Noida, was used for Scanning Electron Microscopy. The Nikon A1 confocal microscope under 40× magnification at AIMT, Amity University, and Uttar Pradesh was used for the confocal study. The flavonoid, terpenoid (LCMSMS-6470 MODEL), chlorophyll, carotenoid (HPLC 1100 SERIES), and saffrole (GCMS-7890(GC), 5975(MS) MODEL) were analyzed.

### 2.3. Fungus Piriformospora indica Culture Conditions

*Piriformospora indica* was grown on 4% jaggery media (Dhampur jaggery) at pH 6.7 with 8–14 days of incubation at 28–30 °C. Similarly, the broth medium was prepared without adding the solidifying agent. Characterization conditions of P. indica spores by SEM-EDX and confocal microscopy SEM (EVO-18, ZEISS, Germany) for SEM analysis and (Oxford instruments, 51-ADD0048) using SMARTSEM software for EDX elemental analysis was carried out to validate the presence of silver in the treated sample.

A culture of *P. indica* spores (control and silver embedded *P. indica* at 300 ppm), grown on 4% jaggery medium plates, was incubated at 28–30 °C for 1 day. The fixation of the specimen was performed in 4% glutaraldehyde for 3 h to preserve its structure and treated for 1 h with 1 mM phosphate-buffered saline (PBS) (pH 7.4) and followed the process by centrifugation for further isolation. After washing with distilled water, the specimen was dehydrated with ascending gradient ethanol series from 20 to 100% EtOH for 10 min each and dried. The specimen was observed under SEM (EVO-18, ZEISS, Germany) for structural analysis, which was carried out at a 10 kV accelerating voltage.

The methodology of Hilbert et al. (2013) with some modifications was followed for confocal microscopy using the Nikon A1 [29]. The control, as well as the AgNPs-treated *P. indica* fungal spores, were gently scrapped from the agar plate using a spreader and cleaned with 0.002% (Tween 20) solution to remove all impurities for confocal microscopy. Then, the samples were collected in a 1 mL centrifuge tube followed by centrifugation at 5000 rpm for 10 min Finally, after washing the pellet with 1 mL autoclaved distilled water, it was observed under 40× magnification with a confocal microscope (Model: Nikon A1, Tokyo, Japan) at AIMT, Amity University, India.

### 2.4. Black Rice Growth Conditions

The NEF formed by *P. indica* treated with chemically synthesized AgNPs showed a significant increase in the spore size, spore count, germination percentage, and biomass in both broth as well as agar media when compared with control *P. indica*. In addition, black rice-treated AgNPs showed positive growth at the 80 ppm-optimized concentration (Figure 1). Further inoculations were performed with the AgNPs, *P. indica*, NEF (both), targeting secondary metabolites production in black rice leaves in pots. 

A pot-based experiment was performed in the greenhouse of the Amity Institute of Microbial Technology. Modified Morishige and Skoog medium was used for in vitro black rice seed germination. After hardening for 15 days, seedlings were transferred to bigger earthen pots (25 cm diameter) containing sand, vermiculite, and sterile soil (1:1:1). The experiment consisted of four treatments, namely (a) untreated black rice (control), (b) only AgNPs-treated (80 ppm) black rice, (c) only *P. indica* (5 × 10^5^ spores mL^−1^)-treated black rice, and (d) Nano-Embedded Fungus (5 × 10^5^ spores mL^−1^)-treated black rice. Whereas the selection of the concentration of AgNPs was optimized for the experiment, the selection of concentration of spores of the *P. indica* was based on a previous study (Dabral et al., 2019). The efficiency of *P. indica* and AgNPs on the growth and secondary metabolite production in black rice was evaluated. Plants were grown for six months under controlled greenhouse conditions (RH 85%; temperature 28 °C, Figure 2).

### 2.5. Secondary Metabolite Analysis

#### 2.5.1. Carotenoids

Carotenoids were extracted using the methodology of Saini (2015) [30]. In total, 1 g of the study specimen was homogenized, of which 3 mL of acetone was taken. Then, 0.1% Butylated hydroxy toluene (BHT) solution in acetone was added to the study specimen as an antioxidant. Buchner’s funnel of 5-micron porosity was used to filter the extract. Washing of the residue with acetone was performed twice until it turned colorless. The filtrate was mixed with 2 gm of anhydrous sodium sulfate. Filtration was performed to remove anhydrous sodium sulfate, and the extract volume was reduced by the Nitrogen evaporator. The extract was moved quantitatively to a 10 mL volumetric flask, and acetone with water was used to ensure the volume reached the mark so that the final extract contains 80% of acetone. The condition of the experiment included an immobile column of Agilent ZORBAX Eclipse Plus C18 with a dimension of 3.0X × 100 mm, Mobile Gradient of 1.8 mm, volume injection of 5 mL, a Detector A red-sensitive FLD, and an excitation of 430 nm/emission of 670 nm. The mobile phase preparation was performed in a mixture of Water–Methanol–Acetonitrile–Dichloromethane (10 + 20 + 70).

#### 2.5.2. Chlorophyll

Chlorophyll concentrations were detected using the methodology of Hornero-Mendez (2005) [31]. In total, 300 mg of black rice leaves were dissolved in petroleum ether (1 mL) and then vortexed. The sample solution was then placed on the cartridge (Phenomenex, Torrance, CA, USA) to rinse it twice with the 1 mL petroleum ether.As the solvent drained to the top of the column packing, the nonpolar substances were quickly eluted with 5 mL petroleum ether/diethyl ether (90:10, *v*/*v*) twice and then discarded. After that, the chlorophyll was eluted with 5 mL of acetone then collected in a glass tube of 10 mL volume and covered with foil. The solvent was evaporated and dried on a rotary evaporator at 20 °C. Then dried residue was then dissolved in another 1 mL acetone for further HPLC analysis. The HPLC conditions were similar to the carotenoid analysis mentioned above.

#### 2.5.3. Saffrole

The homogenized sample (2.0 gm) was taken in a 50 mL polypropylene tube, dissolved in 10 mL MilliQ water, and mixed well. Chloroform (10 mL) was added to the sample and vortexed for 1 min followed by shaking for 30 min on a mechanical shaker. Sodium chloride (3 gm) was added and mixed with vigorous shaking. The sample was further centrifuged for 15 min at 4000 rpm. After centrifugation, the lower layer of chloroform was taken and filtered with a 0.2 μm microfilter and injected into GC-MS [32]. The GC-MS was performed using Agilent GC-7890 with an auto sampler. The column used was HP-5MS UI 15 m, 0.25 μm, 25 mm ID capillary column at 350 °C, with a run time of 12 min. The Inlet program was in split less mode, with 2.7324 mL/min of constant flow with helium gas type. MS conditions included the Agilent 6470 Triple Quad mass spectrometer with 2 filaments at 300 °C temperature and positive-mode electron ionization.

#### 2.5.4. Flavonoids and Terpenoids

Agilent 1290LCMSMS-6470 MODEL was used for the flavonoid and terpenoid analysis following the methodology of Giese (2015) [33]. In total, 50 mg of black rice leaves were extracted in 10 mL of N-Hexane for 24 h in the Soxhlet apparatus. The sample was air dried, extracted with methanol (10 mL), and allowed to dehydrate at 40 °C on a rotatory evaporator. The dehydrated sample was dissolved in methanol (1 mL) and injected into LC-MS/MS. The LC-MS was performed using ESI electron spray ionization mode positive and negative mode at 300 °C ion source temperature with auto sampler. The column ZORBAX RRHD SB C18, 2.1 × 150 mm was used at 30 °C with following conditions: nebulizer pressure 45 psi, drying gas flow rate 9 L/min, fragmentor voltage 250 V, and capillary voltage 3500 V.

### 2.6. Statistical Analysis

The Statistical Package for the Social Sciences Statistics software version 21.0 (SPSS Inc., IBM Corporation, Armonk, NY, USA was used to statistically analyze the data. A one-way analysis of variance (ANOVA) was performed to compare the differences between individual means using Tukey’s honest significant difference (HSD) post-hoc test at *p* ≤ 0.05. All the values are represented as mean of three biological replicates ± standard deviation (SD).

## 3. Results

### 3.1. Characterizations of Silver Nanoparticles

XRD analysis was performed to verify the crystallinity of AgNPs. The obtained XRD spectra showed the XRD peaks 2θ = 38°, 44.14°, 64.24°, 77.24°, 81.36°, 97.66°, respectively, with an intense peak at 2θ = 38°. Figure 3a shows the face-centered cubic (FCC) crystal reflection planes of the four faces of crystalline AgNPs [1,2,16]. XRD peaks confirmed the formation of FCC crystallographic planes which aligned with the standard silver peaks (Figure 3a). UV-Vis absorption spectrophotometry showed a peak around the 400 nm wavelength, characteristic of well-dispersed AgNPs (Figure 3b). The SPR peak of chemically synthesized AgNPs has been obtained at the 400 nm wavelength [1]. Figure 3c shows the Zeta potential of the silver nanoparticles, which was around −30 Mv. This aligns with the standardized Zeta potential readings of chemically synthesized silver nanoparticles.

The FTIR spectrum of silver nanoparticles is shown in Figure 3d. The vibration modes around 3600 cm^−1^ showed the -OH bond vibration. The peak at 2962 and 2926 cm^−1^ were assigned to the symmetric and asymmetric stretching vibrations of -CH bonds in PVP, respectively. The C-N stretch vibration was found at 2187 cm^−1^, and the peaks at 2017, 2068, and 1979, 1739 cm^−1^ were assigned to the -C-C aliphatic stretch and -C-O stretch vibrations, respectively [34]. The symmetric deformation of -CH_3_ in aliphatic compounds was shown by a peak at 1380 cm^−1^. The peaks at 1260 and 1220 cm^−1^ indicate the C-O-C antisymmetric stretch in vinyl compounds. The peak at 1060 cm^−1^ was of the C-O stretch of CH_2_-OH bonds. The peak at 628 cm^−1^ represents the pyridines. All these functional groups of PVP played a vital role in stabilizing the AgNPs [35].

The prepared AgNPs were analyzed using SEM-EDX. The average size of silver nanoparticles varied from 80 to 120 nm. The silver nanoparticles were analyzed using computer image analysis tools, which confirmed hexagonal-shaped silver nanoparticles, as shown in Figure 4. The EDX spectroscopy shows the presence of the Ag element in the sample.

### 3.2. Effects of Silver Nanoparticles on the Growth of P. indica

In the broth medium, the growth of *P. indica* was visibly enhanced by the silver nanoparticles in a concentration-dependent manner (Figure 5). The size and biomass of the colonies increased with the increasing concentration of silver nanoparticles from 100 ppm to 300 ppm and then eventually retarded at 400 ppm as confirmed by fungal colony diameter in agar and dry cell weight in broth. Fungal colony diameters for *P. indica* grown in agar were taken at regular intervals of three days. On the 10th day, the mycelium colony diameter was reported to reach its maximum at 300 ppm AgNPs in comparison to the control (Figure 6).

The increase in (a) fungal colony diameter with an increase in AgNPs concentration in agar and (b) dry cell weight with an increase in AgNPs concentration in broth is seen Figure 7. In comparison to the control (3.6 cm fungal colony diameter) Figure 6(i), maximum growth was observed at 300 ppm (8.4 cm treated colony diameter) Figure 6(ii) in agar after AgNPs treatment and inoculation (Figure 7a). Similarly, the dry cell weight of broth treated with AgNPs at 300 ppm was reported as 6.4 gm (Figure 7b), whereas in the control, it was just 3.8 g. The growth promotion of AgNPs-treated *P. indica* occurred in an increased concentration-dependent method where maximum growth was observed at 300 ppm concentration of AgNPs, as confirmed by the dry cell weight measured after 10 days.

### 3.3. Microscopy Studies

The surface spore and hyphae morphology of *P. indica* using SEM-EDX and confocal microscopy was performed (Figure 8), which showed that in the control, *P. indica spores* were less in number, non-germinating, and small-sized (Figure 8a), whereas those treated with 300 ppm AgNPs were more in number, in the germinating phase with a peculiar germ tube, and larger in size (Figure 8b). The EDX result of the control *P. indica* showed the presence of Carbon (C) and Oxygen (O) (Figure 8c), while the EDX of the AgNPs-treated *P. indica* confirmed the presence of silver along with C and O (Figure 8d). Similar results can be seen in confocal microscopy, as shown in Figure 8e,f, where growth enhancement in *P. indica* spores can be seen. These results confirm the positive growth of *P. indica* spores caused due to the presence of Ag in the treated sample, thereby leading to the formation of NEF.

### 3.4. Secondary Metabolite Analysis in Black Rice Leaves

The optimization of AgNPs in black rice in vitro seed germination was performed as shown in Figure 1, where six different concentrations of AgNPs were taken: control (untreated), 20 ppm, 40 ppm, 60 ppm, 80 ppm, and 100 ppm respectively. Maximum growth was observed at 80 ppm, as seen by the increased root and shoot length after 12 days at 28 °C.

The qualitative and quantitative analysis of secondary metabolites, such as chlorophyll (a and b), carotenoids, flavonoids, terpenoids, and safrole, was carried out in the black rice leaves. Four experimental sets of untreated, only AgNPs-treated, only *P. indica*-treated, and NEF (interaction of both)-optimized *P. indica* with AgNPs were studied in pot culture. The effect NEF on the secondary metabolite composition in black rice (*Oryza sativa* L.) was reported and statistically analyzed (Table 1). Concentrations of majority of secondary compounds were significantly (*p* ≤ 0.05) increased in AgNPs-, *P. indica*-, and NEF-treated plants when compared with the leaves of control plants. Chlorophyll showed an increase of 24.22 %, 58.19 %, and 100% in AgNPs-, *P. indica*-, and NEF-treated plants, respectively. Among the flavonoids, while the concentration of quercetin, apigenin, myricetin, catechin, kaempferol, isorhamnetin, luteolin, and tricine were increased significantly (*p* ≤ 0.05) in all the treatments when compared with the control, the effect of the treatments was non-significant on concentrations of luteolin-7-0-glucosides and isorhamnetin-3-0-glucosides. Terpenoids such as beta cymene, gamma terpinene, terpinene-4-ol, alpha elemene, linalool, caryophyllene, beta ocimene, trans linalool, and myrcene were significantly (*p* ≤ 0.05) increased in AgNPs-, *P. indica*-, and NEF-treated plants. Interestingly, saffrole, a toxic benzodioxole element, was not detected in any of the treatments.

## 4. Discussion

Previous research on the bacterial and viral microbes as an effect of AgNPs on root endophytes is scarce. Here, we formulated nano-embedded *P. indica* that can be utilized to enhance the production of secondary metabolites in black rice (*Oryza sativa* L.) leaves. Morphological changes in the spore and hyphae of *P. indica* after AgNPs treatment were confirmed by SEM and confocal microscopy. The study clearly demonstrated the enhanced fungal growth in response to AgNPs treatment. The interaction of optimized concentrations of AgNPs with NEF showed the production of secondary metabolites in black rice. Similarly, a recent publication with Zn nanoparticles showed enhanced fungal growth as well as enhanced seed germination and root development [36]. Our microscopy analysis also showed that AgNPs enhance the spores as well as hyphae growth. The inoculation of this Nano-Embedded Fungus (NEF) into black rice demonstrated an increased production of secondary metabolites in black rice leaves. For the present research, AgNPs were synthesized, and the impact of fungal endosymbiont was studied by optimizing AgNPs concentration on the interaction with symbiotic mycorrhizal fungus, *P. indica*, which resulted in enhanced productivity of secondary metabolites. As we have observed, the secondary metabolites in black rice leaves showed enormous growth upon being treated with NEF, which shows that NEF has strong growth-promoting effects on the plant system.

Black rice is enriched in anthocyanin pigments, vitamins, proteins, phytochemicals, antioxidants, and secondary metabolites [37]. Some genera in the core microbiome of the black rice plant are well-established plant growth-promoting rhizobacteria PGPRs, which are well known to enhance synthesis of secondary metabolites in plants and improve the antioxidant activity [38,39]. The effective integration of black rice with NEF stimulates the production of secondary metabolites in its leaves. Microscopy analysis also showed that AgNPs enhance the spores as well as hyphae growth of *P. indica*. It was also found that chemically synthesized AgNPs enhanced the growth of fungal spores at 300 ppm concentration in both agar and broth. A positive effect of ZnO nanoparticles on the growth of *Brassica oleracea* var. *botrytis* Broccoli with *P. indica* has also been reported by Singhal et. al (2017) [34]. In addition, ZnO nano materials showed positive growth in medicinal black rice, which may be due to the symplast and apoplast transport channel [18]. The inoculation of the Nano-Embedded Fungus (NEF) in black rice demonstrated positive growth patterns (Figure 2). According to Ferrer et al. (2008), there are about 10,000 flavonoids which have been identified in plants, and their synthesis appears to be pervasive to date. A steady increase of secondary metabolites, namely chlorophyll, carotenoids, flavonoids, and terpenoids in our findings, implies the compositional and essential functional role of endophytic fungal community and its positive correlation or association with AgNPs in black rice, which further contributes to its agricultural implication. The synthesis of active secondary metabolites may be induced due to the endophytic interactions and their hosts [40].

Secondary metabolites are compounds formed by plants through metabolic biosynthetic pathways that make them competitive in their own environment. They induce flowering and fruiting, as well as maintain growth, regulation and signaling in plants. Modern agriculture, as well as the pharmaceutical and nutraceutical sector, depend on plant secondary metabolites for their functioning. According to Vogt et al. (2010), the phenylpropanoid pathway is the pivotal point that initiates various secondary compounds synthesis in the plants [41]. The first enzyme in the phenylpropanoid pathway, phenylalanine ammonium lyase, which turns phenylalanine into p-coumaroyl CoA, which is a key branch point causes the synthesis of plant’s secondary metabolites such as flavonoids, terpenoids, cyanins etc. It can be assumed that *P. indica* along with AgNPs uses a common phenylpropanoid pathway for secondary metabolite synthesis. Khare et al. (2018) reported that an array of common secondary metabolites could be produced from similar precursors in both plants and their endophytes [42]. Our data suggest that the increase of these metabolites in black rice colonized with NEF show strong enhancing and positive effects.

Flavonoids are pivotal drivers of pharmacological action commonly known for their antioxidant activity. Moreover, flavonoids are abundant in plant kingdoms. Quercetin, an important dietary flavonoid, is a potent antioxidant with anti-allergic and anti-inflammatory properties [43]. In the current study, an increase was observed in quercetin levels in plants treated with NEF. Tricin, a potent flavonoid compound from Njavara rice, induces cytotoxic activity and apoptosis against cancer cells. [44] It was also significantly (*p* ≤ 0.05) increased in our study along with apigenin, which possesses insect-resistant properties and can be used as a rice antifeedant. Terpenoids possess antiviral activities and at least 22 terpenoids have been shown to inhibit coronavirus-created havoc by [45]. Linalool, a major monoterpenoid component, is an important scent produced by orchids and was found to be increased by [46]. Therefore, NEF helps in the augmentation of secondary metabolites in black rice. The plant–microbe interaction may be modified to enhance the phytochemical and secondary metabolites production in black rice, but this remains to be explored. More extensive OMICS studies, including genomics, transcriptomics, proteomics, etc., are further needed to study the interactions and functions that occur in black rice. *P. indica* can be grown axenically; therefore, it can be used as a bio-control agent in the agricultural field to overcome the use of chemical fertilizers.

## 5. Conclusions

In the present research work, chemically synthesized AgNPs and their effect on fungal symbiont and black rice was studied by optimizing AgNPs concentration on the interaction with *P. indica*, both the broth as well as the agar, which resulted in an enhanced biomass and increased fungal colony diameter. The optimized AgNPs concentration (300 ppm) interacted with a fungal symbiont called a “Nano-Embedded Fungus”. The inoculation of *P. indica* along with AgNPs treatment in black rice improved growth and productivity. The association of fungal endophyte and AgNPs in correlation with enhancement of secondary metabolites in black rice leaves was described. Our research aims to form a nanotechnology-assisted fungal endosymbiont that could be used as a “bio-formulation” in crop fortification for sustainable agriculture. Hence, it examined the physiological effect of AgNPs on *P. indica* spores and its implication in black rice. Confirmation of its enhanced fungal effect could be useful by integrating it into the agricultural sector.

## Figures and Tables

**Figure 1 jof-09-00260-f001:**
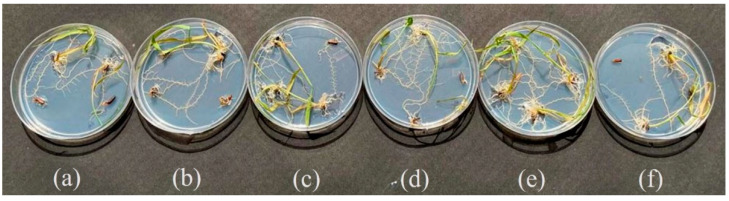
In vitro germination of AgNPs treated black rice seeds showing (**a**) control, (**b**) 20 ppm, (**c**) 40 ppm, (**d**) 60 ppm, (**e**) 80 ppm, and (**f**) 100 ppm.

**Figure 2 jof-09-00260-f002:**
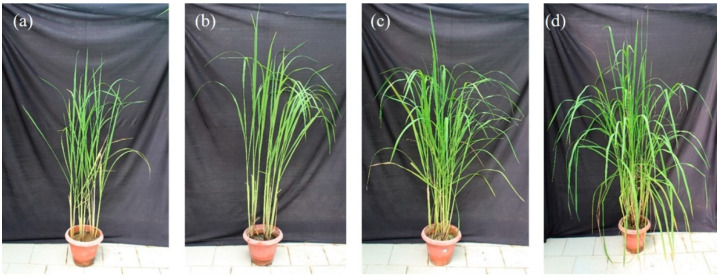
Pot culture experiment of black rice (*Oryza sativa* L.) showing (**a**) control, (**b**) only AgNPs-treated, (**c**) only *P. indica*-treated, and (**d**) NEF (Nano-Embedded Fungus).

**Figure 3 jof-09-00260-f003:**
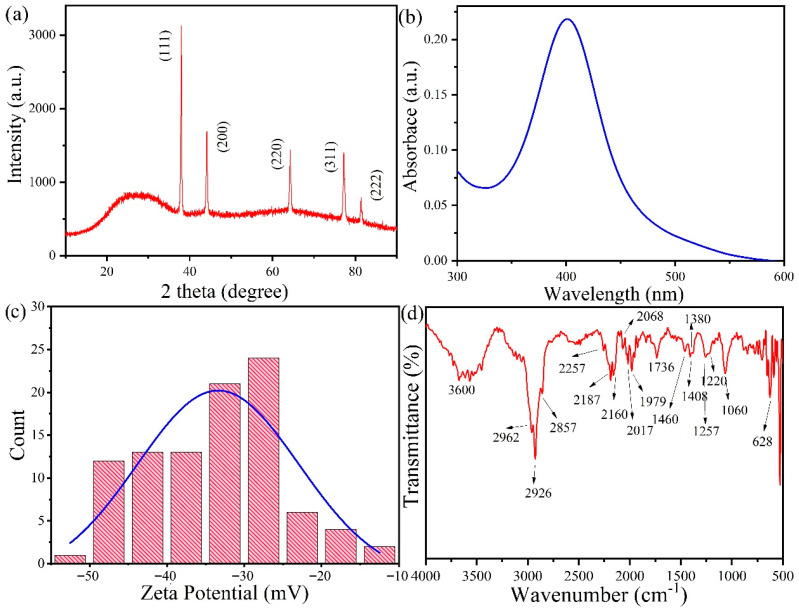
(**a**) X-ray Diffraction Spectroscopy of chemically synthesized silver nanoparticle, (**b**) UV-Vis’s absorption spectrum of silver nanoparticles, (**c**) Zeta potential of chemically synthesized silver nanoparticle, and (**d**) FTIR spectrum scan of chemically synthesized silver nanoparticle.

**Figure 4 jof-09-00260-f004:**
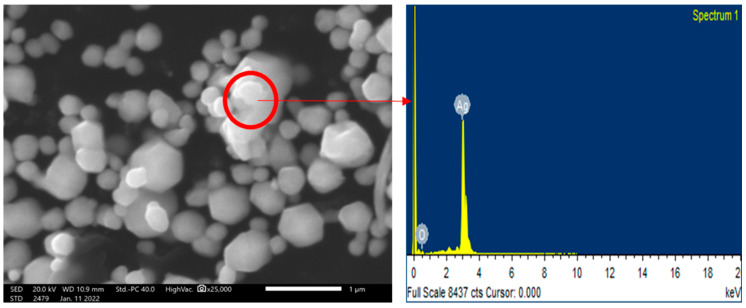
Scanning Electron Micrograph (SEM) of chemically synthesized silver nanoparticles showing hexagonal-shaped nanoparticles, and its EDX spectra showing the Ag element.

**Figure 5 jof-09-00260-f005:**
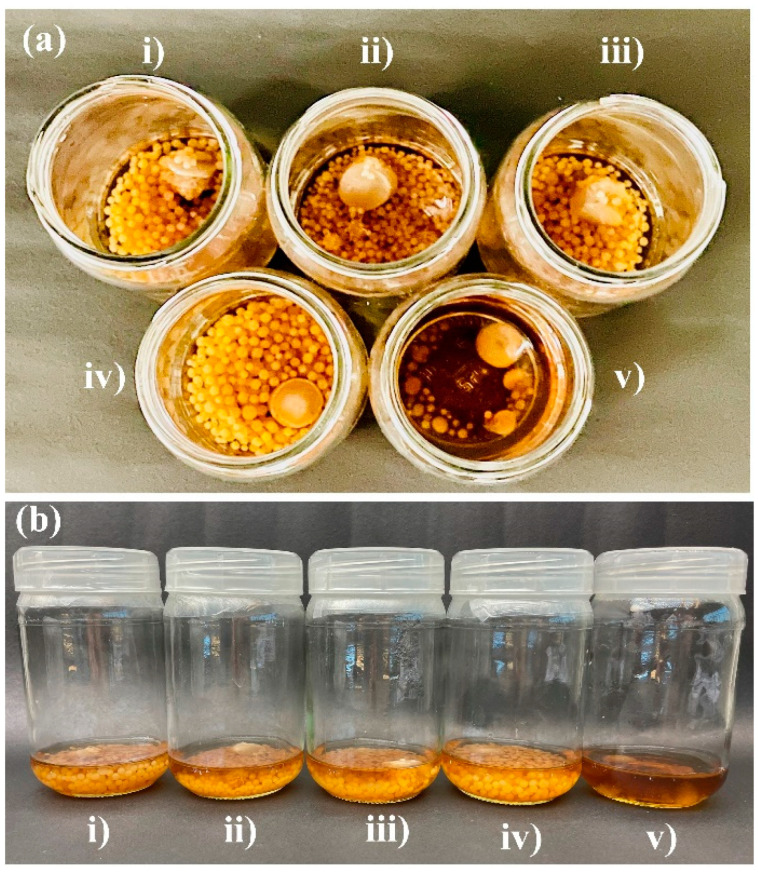
The growth enhancement of *P. indica* when treated with silver nanoparticles in 4% jaggery broth media prepared at pH 6.7: (**a**) Top view, (**b**) front view. Note: (i) Control, (ii) 100 ppm, (iii) 200 ppm, (iv) 300 ppm, and (v) 400 ppm. Growth retardation was observed in higher concentrations (400 ppm) of AgNPs.

**Figure 6 jof-09-00260-f006:**
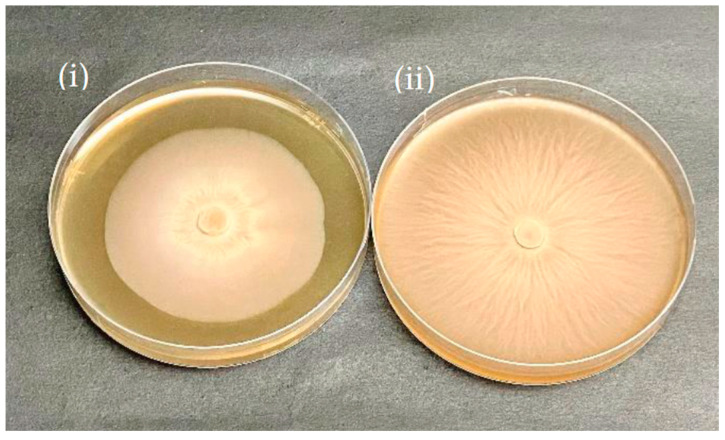
Growth enhancement of silver nanoparticle treated *P. indica* on agar media showing comparative growth enhancement with AgNPs untreated control (i) and treated *P. indica* (ii).

**Figure 7 jof-09-00260-f007:**
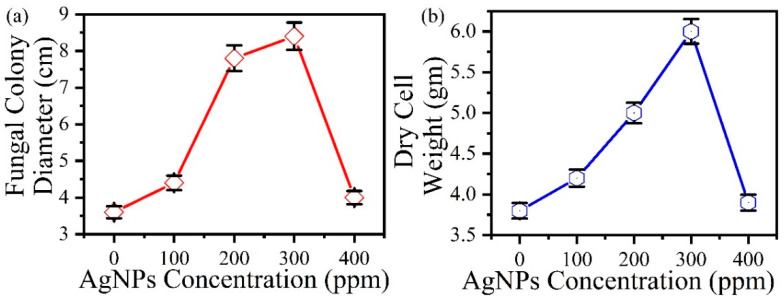
(**a**) The increase in fungal colony diameter with an increase in AgNPs concentration in agar media, (**b**) the increase in dry cell weight with an increase in AgNPs concentration in broth.

**Figure 8 jof-09-00260-f008:**
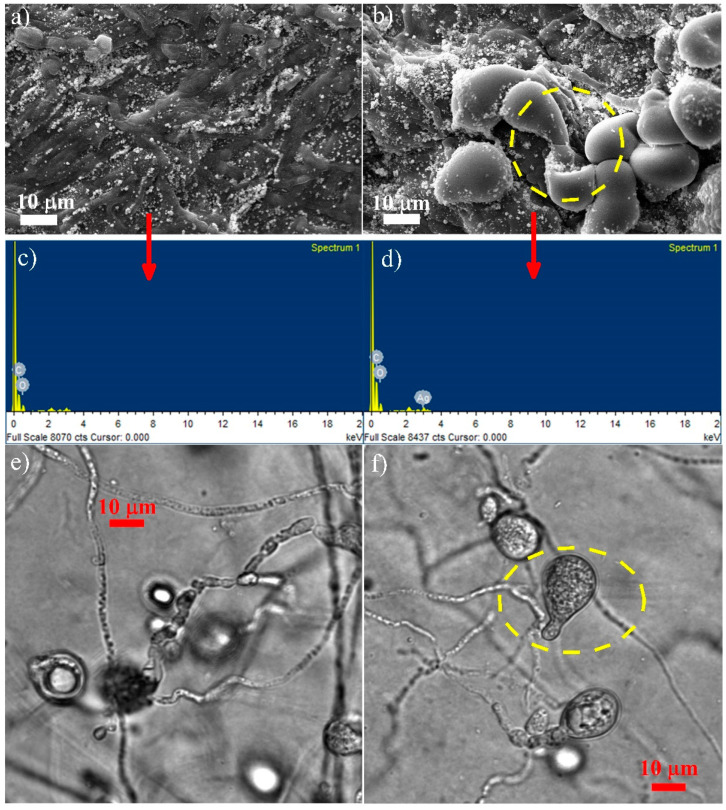
(**a**,**b**) The Scanning Electron Microscopy of the controlled and treated sample, respectively; (**c**,**d**) the EDX graphs of the controlled and treated sample, respectively; (**e**,**f**) the confocal microscopy image of the controlled and treated sample, respectively; evaluating the surface, spores, and hyphae morphology of *P. indica*.

**Table 1 jof-09-00260-t001:** Effect of silver nanoparticle and *Piriformospora indica* (*Serendipita indica*) and their co-application on secondary metabolite composition of black rice *Oryza sativa* L.

Compound	Control (µg/g)	AgNPs (µg/g)	*Piriformospora indica* (µg/g)	NEF (µg/g)
Chlorophyll (a + b)	2928.33 ± 33.93 ^d^	3637.73 ± 42.15 ^c^	4632.38 ± 53.68 ^b^	5852.61 ± 67.82 ^a^
Carotenoid				
Alpha Carotenoid	54.07 ± 0.62 ^d^	71.25 ± 0.82 ^c^	90.31 ± 1.04 ^b^	118.35 ± 1.36 ^a^
Beta Carotenoid	203.71 ± 2.36 ^d^	267.29 ± 3.09 ^c^	343.96 ± 3.98 ^b^	455.69 ± 5.28 ^a^
Safrole	nd	nd	Nd	nd
Flavonoids				
Quercetin	177.85 ± 2.05 ^d^	234.77 ± 2.72 ^c^	297 ± 3.44 ^b^	421.75 ± 4.88 ^a^
Apigenin	237.58 ± 2.75 ^d^	313.60 ± 3.63 ^c^	397.77 ± 4.61 ^b^	564.84 ± 6.54 ^a^
Myricetin	199.51 ± 2.31 ^d^	262.35 ± 3.04 ^c^	334.18 ± 3.87 ^b^	474.55 ± 5.5 ^a^
Catechin	208.53 ± 2.4 ^d^	275.26 ± 3.16 ^c^	352.92 ± 4.09 ^b^	494.53 ± 5.69 ^a^
Kaempferol	138.17 ± 1.58 ^d^	187.34 ± 3.71 ^c^	235.86 ± 2.73 ^b^	339.46 ± 6.72 ^a^
Isorhamnetin	183.61 ± 3.63 ^d^	242.43 ± 4.8 ^c^	302.54 ± 3.5 ^b^	435.41 ± 8.62 ^a^
Luteolin	259.99 ± 5.14 ^d^	343.20 ± 6.79 ^c^	427.40 ± 4.95 ^b^	606.91 ± 7.03 ^a^
Luteolin-7-0-glucosides	71.56 ± 1.42 ^a^	83.52 ± 11.14 ^a^	71.48 ± 7.55 ^a^	76.48 ± 3.19 ^a^
Apigenin-7-0-glucoside	61.66 ± 0.57 ^a^	56.68 ± 0.60 ^b^	56.34 ± 1.06 ^b^	61.96 ± 2.70 ^a^
Quercetin-3-0-rutinosides	51.83 ± 1.03 ^a^	51.05 ± 0.94 ^a^	45.78 ± 0.22 ^b^	48.35 ± 0.75 ^b^
Isorhamnetin-3-0-glucosides	67.28 ± 0.75 ^a^	66.14 ± 1.08 ^a^	63.54 ± 1.60 ^a^	63.20 ± 2.46 ^a^
Quercetin-3-0-glucosides	73.61 ± 1.44 ^a^	63.49 ± 0.87 ^c^	67.77 ± 1.19 ^b^	66.72 ± 1.40 ^bc^
Tricin	1330.87 ± 26.35 ^d^	1757.78 ± 34.81 ^c^	2222.56 ± 44.01 ^b^	3156.05 ± 62.49 ^a^
Tricin-4-0 erythro-B guaiacylglyceryl	59.27 ± 0.87 ^b^	59.25 ± 1.11 ^b^	60.52 ± 1.63 ^b^	65.48 ± 1.42 ^a^
Terpenoids				
Beta Cymene	370.52 ± 7.33 ^d^	479.25 ± 1.03 ^c^	610.13 ± 1.00 ^b^	821.66 ± 16.27 ^a^
Gamma-Terpinene	99.21 ± 2 ^d^	130.99 ± 2.64 ^c^	173.48 ± 3.43 ^b^	229 ± 4.53 ^a^
Terpinen-4-ol	135.69 ± 2.68 ^d^	179.18 ± 3.55 ^c^	227.97 ± 4.51 ^b^	300.92 ± 5.96 ^a^
Beta-Pinene	47.97 ± 0.95 ^c^	64.25 ± 1.27 ^ab^	80.6 ± 1.59 ^ab^	139.68 ± 59.49 ^a^
Alpha-Elemene	121.56 ± 2.46 ^d^	160.39 ± 3.24 ^c^	209.75 ± 2.40 ^b^	281.63 ± 5.57 ^a^
Linalool	53.06 ± 1.05 ^d^	70.14 ± 1.39 ^c^	89.15 ± 1.76 ^b^	117.88 ± 2.33 ^a^
10-acetylmethyl-3-carene	97.96 ± 1.94 ^d^	129.41 ± 2.56 ^c^	164.58 ± 3.26 ^b^	217.27 ± 4.30 ^a^
Carveol	13.74 ± 0.28 ^c^	15.73 ± 1.21 ^bc^	17.32 ± 1.14 ^b^	31.73 ± 0.63 ^a^
Limonene	308.74 ± 6.11 ^d^	407.79 ± 8.07 ^c^	518.8 ± 10.27 ^b^	684.66 ± 13.55 ^a^
Caryophyllene	70.41 ± 1.39 ^d^	93.37 ± 1.85 ^c^	118.3 ± 2.34 ^b^	156.17 ± 3.09 ^a^
Sabinene	70.23 ± 1.41 ^b^	68.03 ± 1.78 ^b^	70.68 ± 1.14 ^b^	157.32 ± 1.55 ^a^
Beta-Ocimene	202 ± 4 ^d^	266.91 ± 5.28 ^c^	339.46 ± 6.72 ^b^	447.97 ± 8.87 ^a^
Trans-Linalool	135.89 ± 2.69 ^d^	179.40 ± 3.55 ^c^	228.29 ± 4.52 ^b^	302.62 ± 5.99 ^a^
Cis linalool	59.81 ± 1.21 ^b^	69.91 ± 3.40 ^b^	66.24 ± 3.80 ^b^	134.99 ± 3.92 ^a^
Myrcene	198.95 ± 3.93 ^d^	232.06 ± 4.59 ^c^	334.24 ± 6.61 ^b^	442.60 ± 8.76 ^a^

Values represent means of three biological replicates ± SD. Different letters within each column represent significant difference among the treatments at *p* ≤ 0.05, derived from Tukey’s HSD. AgNPs—silver nanoparticles; NEF—Nano-Embedded Fungi; nd—not detected.

## Data Availability

Not applicable.

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
