# Peer review of "Co-Application of Silver Nanoparticles and Symbiotic Fungus Piriformospora indica Improves Secondary Metabolite Production in Black Rice"

_jof, 2023, doi:10.3390/jof9020260_

Round 1

Reviewer 1 Report

The study “Co-application of silver nanoparticles and symbiotic fungus Piriformospora indica improves secondary metabolite production in black rice” by Varma et al., reported the synergistic effect of AgNPs and the fungal symbionts in augmenting the secondary metabolites in leaves of black rice. The application of Nano-Embedded Fungus (NEF) caused increase in fungal biomass, colony diameter, spore count, and spore size than the control P. indica. Treatment with AgNPs, P. indica and NEF resulted in growth enhancement in black rice. The study is interesting in somewhat presenting new information on the investigated topic. However, several points need clarification. Find the below comments for improvements.

Keywords should be different from title.

The NPs synthesis was not an objective of this study. The method and chemical for the synthesis of NPs is already reported. Commercially available NPs can be used or the NPs should be investigated using non-chemical green approaches.

Line 66: one of the aim of this study was to evaluate the effect of applying NEF on the productivity of crop. But I didn’t find the data related to production or growth attributes

Please provide the source of fungus. From where it was isolated? How identified ?? etc.

Line 147: “Experiments colonized by P. indica treated with chemically”, experiment was not colonized by fungus. Rephrase this and some other sentences at several places to make the meaning clearer

What was the spore concentration of the fungus at the time of application?

Data in table 1 should be analyzed statistically to see the significant difference among treatments and control

Discussion is diverging in several directions also in some that are not investigated in this study. Please discuss the results specifically obtained in this study

Author Response

Reviewer 1

  1. Keywords should be different from title.

Ans. Thank you for your valuable comment. As suggested by the reviewer, the keywords in the revised manuscript have been changed.

Serendipita indica, Endosymbiont, Nano-Embedded Fungus, Confocal microscopy, Scanning Electron Microscopy, Oryza sativa L. indica, Nano-Bioformulation”.

  1. The NPs synthesis was not an objective of this study. The method and chemical for the synthesis of NPs is already reported. Commercially available NPs can be used or the NPs should be investigated using non-chemical green approaches.

Ans. Thank you for your valuable comment. We tried synthesizing AgNPs using green approaches but could not achieve the purity required for our analysis. We are in the process of refining the green approach which we will report in future publications. Same problem was observed with the commercially available NPs. Therefore, we have synthesized the AgNPs using chemical approach as this method can create highly pure and reproducible AgNPs in large quantity which was required for the current study.

  1. Line 66: one of the aim of this study was to evaluate the effect of applying NEF on the productivity of crop. But I didn't find the data related to production or growth attributes

Ans. Thank you for pointing out the issue in line 66. The present study is mainly focused on addressing the effect of NEF on the secondary metabolites of black rice under different conditions and was limited to the evaluation at the leaf stage. Therefore, parameters such as grain yield, grain biomass and other agronomic features were not assessed initially. As suggested by the esteemed reviewer on providing data on crop productivity, we are working on the manuscript that focuses on the crop productivity and nutritional aspects.

 For better clarity, we have rephrased line 66-68 as “This research aims to evolve a nanotechnology-augmented fungal endo-symbiont in order to enhance the production of secondary metabolites in black rice (Oryza sativa L.) leaves.” in the revised manuscript.

  1. Please provide the source of fungus. From where it was isolated? How identified 2? Etc

Ans. Thank you for your valuable comment. “Piriformospora indica is an endophytic root-colonising species discovered, screened, and identified from orchid plants of the Thar deserts in Rajasthan, India by Prof Ajit Varma and group, Jawaharlal Nehru University, New Delhi (Verma et al., 2018) and maintained at Amity Institute of Microbial Technology (AIMT) as accession number DSM 11827. The pure inoculum was obtained from the (AIMT) laboratory and was maintained in 4% jaggery media following the methodology of Uma et al., (2017).”

This has now been incorporated in lines 87-92 of the revised manuscript.

  1. Line 147: "Experiments colonized by P. indica treated with chemically" experiment was not colonized by fungus. Rephrase this and some other sentences at several places to make the meaning clearer

Ans. Thank you for your valuable comment. The line has been rephrased in lines 150-152 for better clarity in the revised manuscript.

“The NEF formed by P. indica treated with chemically synthesized AgNPs showed a significant increase in the spore size, spore count, germination percentage, and biomass in both broth as well as agar media when compared with control P. indica.

  1. What was the spore concentration of the fungus at the time of application?

Ans. Thank you for your valuable comment. The concentration of the P. indica spores used at the time of application was 5 × 105 spores mL-1. As suggested by the reviewer, this has now been mentioned in lines 165-169 of the revised manuscript.

“After hardening for 15 days, seedlings were transferred to bigger earthen 162 pots (25 cm diameter) containing sand, vermiculite, and sterile soil (1:1:1). The experiment consisted of four treatments namely, (a) Untreated black rice (Control) (b) only AgNPs treated (80 ppm) black rice (c) only P. indica (5 × 105 spores mL-1) treated black rice (d) Nano-Embedded Fungus (5 × 105 spores mL-1) treated black rice. While the selection of the concentration of AgNPs was optimized for the experiment, the selection of concentration of spores of the P. indica was based on the previous study (Dabral et al., 2019).”

  1. Data in table 1 should be analyzed statistically to see the significant difference among treatments and control

Ans. Thank you for your valuable comment. As suggested by the esteemed reviewer, the statistical analysis was done using Statistical Package for the Social Sciences Statistics software version 21.0 (SPSS Inc., IBM Corporation). One-way analysis of variance (ANOVA) was done for comparing the differences between individual means using Tukey’s honestly significant difference (HSD) post hoc test at p≤ 0.05 . All the values were represented as means of three biological replicates ± standard deviation (SD).

The statistical results have now been incorporated in the results and table 1 of the revised manuscript.

 Table 1. Effect of silver nanoparticle and Piriformospora indica (Serendipita indica) and their co-application on secondary metabolite composition of black rice Oryza sativa L.

Compound

Control

(µg/g)

AgNPs

(µg/g)

Piriformospora indica (µg/g)

NEF

(µg/g)

Chlorophyll (a+b)

2928.33±33.93 d

3637.73±42.15 c

4632.38±53.68 b

5852.61±67.82a

Carotenoid

Alpha Carotenoid

54.07±0.62 d

71.25±0.82 c

90.31±1.04 b

118.35±1.36 a

Beta Carotenoid

203.71±2.36 d

267.29±3.09 c

343.96±3.98 b

455.69±5.28 a

Safrole

nd

nd

nd

nd

Flavonoids

Quercetin

177.85±2.05 d

234.77±2.72 c

297±3.44 b

421.75±4.88 a

Apigenin

237.58±2.75 d

313.60±3.63 c

397.77±4.61 b

564.84±6.54 a

Myricetin

199.51±2.31 d

262.35±3.04 c

334.18±3.87 b

474.55±5.5 a

Catechin

208.53±2.4 d

275.26±3.16 c

352.92±4.09 b

494.53±5.69 a

Kaempferol

138.17±1.58 d

187.34±3.71 c

235.86±2.73 b

339.46±6.72 a

Isorhamnetin

183.61±3.63 d

242.43±4.8 c

302.54±3.5 b

435.41±8.62 a

Luteolin

259.99±5.14 d

343.20±6.79 c

427.40±4.95 b

606.91±7.03 a

Luteolin-7-0-glucosides

71.56±1.42 a

83.52±11.14 a

71.48±7.55 a

76.48±3.19 a

Apigenin-7-0-glucoside

61.66±0.57 a

56.68±0.60 b

56.34±1.06 b

61.96±2.70 a

Quercetin-3-0-rutinosides

51.83±1.03a

51.05±0.94 a

45.78±0.22 b

48.35±0.75 b

Isorhamnetin-3-0-glucosides

67.28±0.75 a

66.14±1.08 a

63.54±1.60 a

63.20±2.46 a

Quercetin-3-0-glucosides

73.61±1.44 a

63.49±0.87 c

67.77±1.19 b

66.72±1.40 bc

Tricin

1330.87±26.35 d

1757.78±34.81 c

2222.56±44.01 b

3156.05±62.49 a

Tricin-4-0 erythro-B guaiacylglyceryl

  59.27±0.87 b

59.25±1.11 b

60.52±1.63 b

65.48±1.42 a

Terpenoids

Beta Cymene

370.52±7.33d

479.25±1.03 c

610.13±1.00 b

821.66±16.27 a

Gamma-Terpinene

99.21±2 d

130.99±2.64 c

173.48±3.43 b

229±4.53 a

Terpinen-4-ol

135.69±2.68 d

179.18±3.55 c

227.97±4.51 b

300.92±5.96 a

Beta-Pinene

47.97±0.95 c

64.25±1.27 ab

80.6±1.59 ab

139.68±59.49 a

Alpha-Elemene

121.56±2.46 d

160.39±3.24 c

209.75±2.40 b

281.63±5.57 a

Linalool

53.06±1.05 d

70.14±1.39 c

89.15±1.76 b

117.88±2.33 a

10-acetylmethyl-3-carene

97.96±1.94 d

129.41±2.56 c

164.58±3.26 b

217.27±4.30 a

Carveol

13.74±0.28 c

      15.73±1.21 bc

17.32±1.14 b

31.73±0.63 a

Limonene

308.74±6.11 d

407.79±8.07 c

518.8±10.27 b

684.66±13.55 a

Caryophyllene

70.41±1.39 d

93.37±1.85 c

118.3±2.34 b

156.17±3.09 a

Sabinene

70.23±1.41 b

68.03±1.78 b

70.68±1.14 b

157.32±1.55 a

Beta-Ocimene

202±4 d

266.91±5.28 c

339.46±6.72 b

447.97±8.87 a

Trans-Linalool

135.89±2.69 d

179.40±3.55 c

228.29±4.52 b

302.62±5.99 a

Cis linalool

59.81±1.21 b

69.91±3.40 b

66.24±3.80 b

134.99±3.92 a

Myrcene

198.95±3.93 d

232.06±4.59 c

334.24±6.61 b

442.60±8.76 a

Values represent means of three biological replicates ± SD. Different letters within each column represent significant difference among the treatments at p≤0.05, derived from Tukey HSD.

AgNPs- silver nanoparticles; NEF- nano embedded fungi; nd- not detected

  1. Discussion is diverging in several direction also in some that are not investigated in this study. Please explain the results specifically obtained in this study.

Ans. Thank you for your valuable comment. Done as suggested.

Reviewer 2 Report

Piriformospora indica is an important representative of endophytic fungi. 

This paper presents a novel method of synergistic promotion of rice growth by nanoparticles and endophytic fungi.

It provides a scientific basis for the subsequent application in the field.

There are a few minor suggestions, as follows:

1. In Figure 5, please indicate the standard deviation.

2. The pixel of Figure 6 is low. Please raise the pixel. And the scale bar in Fig 6 e, f is unclear.

3. In Table 1, please note significance.

4. Please mark Table 1 in section 3.4.

Author Response

Reviewer 2

Comments and questions for Authors

Pinformospora indica is an important representative of endophytic fungi.

This paper presents a novel method of synergistic promotion of rice growth by nanoparticles and endophytic fungi.

It provides a scientific basis for the subsequent application in the field.

There are a few minor suggestions, as follows.

  1. In Figure 5, please indicate the standard deviation.

Ans. Thank you for your valuable comment. We have added the standard deviation in Figure 5.

  1. The pixel of Figure 6 is low. Please raise the pixel. And the scale bar in Fig 6 e, fig unclear.

Ans. Thank you for your valuable comment. As suggested by the esteemed reviewer, the pixels and resolution of the figure 6 have been improved. A clear scale bar is also incorporated in figure 6 of the revised manuscript.

  1. In Table 1, please note significance.

Ans. Thank you for your valuable comment. The data in table 1 has now been statistically analysed. One-way analysis of variance (ANOVA) was done for comparing the differences between individual means using Tukey’s honestly significant difference (HSD) post hoc test at p≤ 0.05. All the values were represented as means of three biological replicates ± standard deviation (SD).

Table 1. Effect of silver nanoparticle and Piriformospora indica (Serendipita indica) and their co-application on secondary metabolite composition of black rice Oryza sativa L.

Compound

Control

(µg/g)

AgNPs

(µg/g)

Piriformospora indica (µg/g)

NEF

(µg/g)

Chlorophyll (a+b)

2928.33±33.93 d

3637.73±42.15 c

4632.38±53.68 b

5852.61±67.82a

Carotenoid

Alpha Carotenoid

54.07±0.62 d

71.25±0.82 c

90.31±1.04 b

118.35±1.36 a

Beta Carotenoid

203.71±2.36 d

267.29±3.09 c

343.96±3.98 b

455.69±5.28 a

Safrole

nd

nd

nd

nd

Flavonoids

Quercetin

177.85±2.05 d

234.77±2.72 c

297±3.44 b

421.75±4.88 a

Apigenin

237.58±2.75 d

313.60±3.63 c

397.77±4.61 b

564.84±6.54 a

Myricetin

199.51±2.31 d

262.35±3.04 c

334.18±3.87 b

474.55±5.5 a

Catechin

208.53±2.4 d

275.26±3.16 c

352.92±4.09 b

494.53±5.69 a

Kaempferol

138.17±1.58 d

187.34±3.71 c

235.86±2.73 b

339.46±6.72 a

Isorhamnetin

183.61±3.63 d

242.43±4.8 c

302.54±3.5 b

435.41±8.62 a

Luteolin

259.99±5.14 d

343.20±6.79 c

427.40±4.95 b

606.91±7.03 a

Luteolin-7-0-glucosides

71.56±1.42 a

83.52±11.14 a

71.48±7.55 a

76.48±3.19 a

Apigenin-7-0-glucoside

61.66±0.57 a

56.68±0.60 b

56.34±1.06 b

61.96±2.70 a

Quercetin-3-0-rutinosides

51.83±1.03a

51.05±0.94 a

45.78±0.22 b

48.35±0.75 b

Isorhamnetin-3-0-glucosides

67.28±0.75 a

66.14±1.08 a

63.54±1.60 a

63.20±2.46 a

Quercetin-3-0-glucosides

73.61±1.44 a

63.49±0.87 c

67.77±1.19 b

66.72±1.40 bc

Tricin

1330.87±26.35 d

1757.78±34.81 c

2222.56±44.01 b

3156.05±62.49 a

Tricin-4-0 erythro-B guaiacylglyceryl

  59.27±0.87 b

59.25±1.11 b

60.52±1.63 b

65.48±1.42 a

Terpenoids

Beta Cymene

370.52±7.33d

479.25±1.03 c

610.13±1.00 b

821.66±16.27 a

Gamma-Terpinene

99.21±2 d

130.99±2.64 c

173.48±3.43 b

229±4.53 a

Terpinen-4-ol

135.69±2.68 d

179.18±3.55 c

227.97±4.51 b

300.92±5.96 a

Beta-Pinene

47.97±0.95 c

64.25±1.27 ab

80.6±1.59 ab

139.68±59.49 a

Alpha-Elemene

121.56±2.46 d

160.39±3.24 c

209.75±2.40 b

281.63±5.57 a

Linalool

53.06±1.05 d

70.14±1.39 c

89.15±1.76 b

117.88±2.33 a

10-acetylmethyl-3-carene

97.96±1.94 d

129.41±2.56 c

164.58±3.26 b

217.27±4.30 a

Carveol

13.74±0.28 c

      15.73±1.21 bc

17.32±1.14 b

31.73±0.63 a

Limonene

308.74±6.11 d

407.79±8.07 c

518.8±10.27 b

684.66±13.55 a

Caryophyllene

70.41±1.39 d

93.37±1.85 c

118.3±2.34 b

156.17±3.09 a

Sabinene

70.23±1.41 b

68.03±1.78 b

70.68±1.14 b

157.32±1.55 a

Beta-Ocimene

202±4 d

266.91±5.28 c

339.46±6.72 b

447.97±8.87 a

Trans-Linalool

135.89±2.69 d

179.40±3.55 c

228.29±4.52 b

302.62±5.99 a

Cis linalool

59.81±1.21 b

69.91±3.40 b

66.24±3.80 b

134.99±3.92 a

Myrcene

198.95±3.93 d

232.06±4.59 c

334.24±6.61 b

442.60±8.76 a

Values represent means of three biological replicates ± SD. Different letters within each column represent significant difference among the treatments at p≤0.05, derived from Tukey HSD.

AgNPs- silver nanoparticles; NEF- nano embedded fungi; nd- not detected

  1. Please mark Table 1 in section 3.4

Ans. Thank you for your valuable comment. We have made the correction as suggested in the revised manuscript.

Round 2

Reviewer 1 Report

Authors have improved the manuscript according to comments.